# Study on Residual Stress of Welded Hoop Structure

**Wenbo Ma [1],\*, Heng Zhang [1], Wei Zhu [2], Fu Xu [1] and Caiqian Yang [1],\***

[1] College of Civil Engineering and Mechanics, Xiangtan University, Xiangtan 410075, China; zhangheng940610@163.com (H.Z.); xufu@xtu.edu.cn (F.X.)

[2] School of Mathematics and Computational Science, Xiangtan University, Xiangtan 410075, China; zhuwei@xtu.edu.cn

\* Correspondence: mawenbo@xtu.edu.cn (W.M.); ycqjxx@seu.edu.cn (C.Y.); Tel.: +86-1587-483-1227 (W.M.); +86-1896-361-2628 (C.Y.)

**Abstract:** Residual stress is inevitable during welding, which will greatly affect the reliability of the structure. The purpose of this paper was to study the residual stress of the hoop structure caused by the cooling shrinkage of the weld when the outer cylinder was wrapped and welded under the condition of the existing inner cylinder. In this paper, the "method of killing activating elements" of ANSYS was used to simulate the three-dimensional finite element of the hoop structure. In the case of applying interlayer friction, the welding-forming process and welding circumferential residual stress of the hoop structure were analyzed. The blind hole method was used to test the residual stress distribution of the hoop structure, and the test results were compared with the finite element simulation results to verify the reliability of the simulation calculation method and the reliability of the calculation results. Then, the influence factors of the maximum welding residual stress of the hoop structure were studied. The results show that the maximum residual stress of the outer plate surface of the hoop structure decreases with the increase of the welding energy, the thickness of the laminate, the width of the weld seam, the welding speed, and the radius of the container. Based on the results of numerical simulation, the ternary first-order equations of the maximum residual stress of the hoop structure with respect to the welding speed, the thickness of the laminate, and the width of the weld seam were established. Then, the optimal welding parameters were obtained by optimizing the equations, which provided an important basis for the safe use and optimal design of the welding hoop structure.

**Keywords:** welding; hoop structure; numerical simulation; finite element model; residual stress

## 1. Introduction

The layered cylindrical vessel has wide engineering applications [1–3]. It can be divided into several categories depending on the layered vessel type, such as concentric wrapped shell, coil-wound, shrink fit, and spiral wrapped [4,5]. It is a common problem to crack propagation in the circumferential weld of a layered cylindrical vessel. It is easy for such cracks to cause structural failure when they are subjected to local stress caused by an external load, thermal stress, and residual stress [6–9]. Therefore, it is necessary to predict residual welding stress and evaluate it effectively [10].

The welding hoop structure is a kind of constriction structure produced by welding shrinkage in the layered cylindrical vessel during layered binding welding. This tightening structure will generate a large circumferential residual tensile stress on the laminate, especially around the weld. This kind of residual stress greatly increases the risk of cracks in the weld joints and reduces the reliability of the welded structure. However, for large weldments, it is complicated to carry out an overall tempering treatment, and a large number of residual stresses are difficult to be eliminated [11,12].

Therefore, it is very important to study the residual stress and optimize the welding process of the welding hoop structure.

Combining the experimental measurement with numerical simulation is the most effective solution to investigate the residual welding stress [13]. In recent years, the finite element method has been proved to be useful and compelling to evaluate the welding temperature field, residual stress field, and the effect of post-weld heat treatment [14–21]. There is much software that can perform finite element simulation of the welding process, such as SYSWELD, Simufact Welding, ANSYS, ABAQUS, etc. They can all simulate the welding process very well [22–24]. At the same time, software such as ANSYS can also simulate the multilayer structure very well [25]. In the case of layered pressure vessels, Xu and Wang et al. improved the welding structure of the layered urea reactor and evaluated the modified structure by ANSYS software [26]. Then, they used ABAQUS to predict the residual stress in layer-to-layer joints [27]. Xu and We et al. used ABAQUS to simulate the welding temperature and predict residual stress in welding and the heat-affected zone between nozzle and nozzle [28]. Xu and Wang et al. used ABAQUS to study the distribution of residual stress in the weld and heat-affected zone of the delamination joint with the cladding layer [29]. However, their research ignored the characteristics of the hoop structure.

According to the actual manufacturing process and structural characteristics of the wrapped pressure vessel, the inner and outer surfaces of the inner cylinder and the laminate cannot be processed into absolutely smooth surfaces, but instead surfaces with a certain roughness. Therefore, during the bandaging process, whether under the action of the bandaging force or when the longitudinal seam of the laminate cools and contracts, there is interlayer friction in the relative movement of the laminate [30]. In many welding residual stress studies, the simplification of the numerical model generally ignored the characteristics of the interlayer friction and the clamping structure, which would inevitably lead to the gap between the simulation results and the actual data. It could not provide the correct theoretical basis for the analysis and prediction of residual welding stress. Therefore, the analysis of the residual stress distribution rule of the welding hoop structure and the influence of the welding process on the residual welding stress is of great significance for the formulation of the correct welding process, the improvement of the reliability of the welding structure, and the extension of the equipment life. In this paper, a banded pressure vessel was used as the modeling prototype, and ANSYS was used to carry out a three-dimensional finite element numerical simulation of the hoop structure. The welding residual stress distribution trend was studied. The simulation results were compared with the welding test results under the same process conditions, and the reliability of the simulation calculation method and the correctness of the calculation results were verified. Then, the influence factors of the model's maximum welding residual stress were studied to provide a basis for the safe use and optimal design of the welding hoop structure.

## 2. Finite Element Model

### 2.1. Material Physical Properties

The change of thermophysical properties and mechanical properties of materials with temperature will significantly affect the accuracy of the welding simulation calculation results. Therefore, a bilinear follow-up elastoplastic constitutive model is established to simulate the change of elastic modulus caused by metal exciting heat and cooling during welding. It is assumed in the numerical simulation that all materials conform to the Mises yield criterion and the bilinear follow-up hardening model [31,32]. The constitutive relationship parameters are shown in Table 1. The thermophysical performance parameters are shown in Table 2 [33].

**Table 1.** The parameters of the constitutive relation of bilinear dynamic elastic–plastic.

| Temperature $T$ (°C) | Elastic Modulus before Yield $E$ (GPa) | Yield Stress $\sigma$ (MPa) | Elastic Modulus after Yielding $E'$ (GPa) |
|---|---|---|---|
| 20 | 203 | 356 | 20.3 |
| 100 | 194 | 321 | 19.4 |
| 200 | 188 | 321 | 18.8 |
| 300 | 186 | 312 | 18.6 |
| 400 | 174 | 283 | 17.4 |
| 500 | 161 | 249 | 16.1 |
| 600 | 136 | 170 | 13.6 |
| 1000 | 10 | 11 | 1 |
| 1500 | 10 | 5 | 1 |
| 2000 | 1 | 0.5 | 0.1 |

**Table 2.** Thermal–physical properties parameters of Q345R steel.

| Temperature $T$ (°C) | Density $\rho$ (kg·m$^{-3}$) | Enthalpy $h$ (J·m$^{-3}$) | Thermal Conductivity $\lambda$ (W·m·°C$^{-1}$) | Thermal Expansion Coefficient $\alpha$ (W·m$^{-2}$·K$^{-1}$) | Specific Heat Capacity C (J·Kg·°C$^{-1}$) | Poisson's Ratio $\mu$ |
|---|---|---|---|---|---|---|
| 0 | 7800 | — | 18.9 | $1.10 \times 10^{-5}$ | 580 | 0.35 |
| 400 | 7550 | 1664 | 28.35 | $1.36 \times 10^{-5}$ | 756 | 0.35 |
| 600 | 7550 | 2779 | 31.08 | $1.39 \times 10^{-5}$ | 840 | 0.35 |
| 800 | 7550 | 3406 | 32.76 | $1.48 \times 10^{-5}$ | 882 | 0.35 |
| 1000 | 7550 | 4079 | 31.5 | $1.34 \times 10^{-5}$ | 756 | 0.35 |
| 1500 | 7550 | 4797 | 28.5 | $1.33 \times 10^{-5}$ | 797 | 0.35 |

## 2.2. Model Establishment

Using ANSYS software, a banded pressure vessel (Figure 1) was used as a prototype to carry out numerical simulation calculations on the welding process of the double-layer hoop structure. The inner diameter of the structure is R. The welding joint adopts a V-shaped groove, and the angle of the weld groove is 40°. The thickness of the weld (the thickness of the outer plate) is b, and the weld width is δ. Based on the actual welding of the layered pressure vessels, displacement constraints were set at the bottom and left side of the hoop structure model. Set the interlayer friction coefficient to 0.6 to simulate the interlayer friction. The coupling element SOLID5 with temperature and displacement degrees of freedom was selected to perform thermal–structural coupling analysis of the welding process by the direct method. In the element mesh division, considering that there are significant temperature and stress gradients in the weld area and the near-weld area, the mesh was densified so that the mesh size of the weld and its surrounding area is not more than 2 mm. The mesh division results are shown in Figure 2. The gray mesh area in the figure is the weld joint.

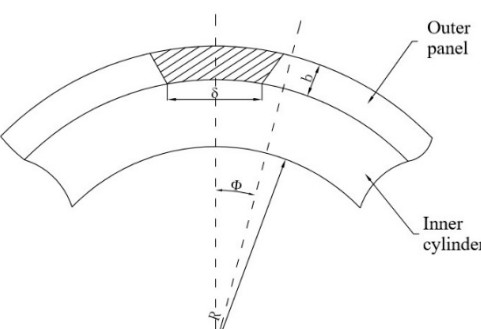

**Figure 1.** Schematic diagram of a bandaged pressure vessel.

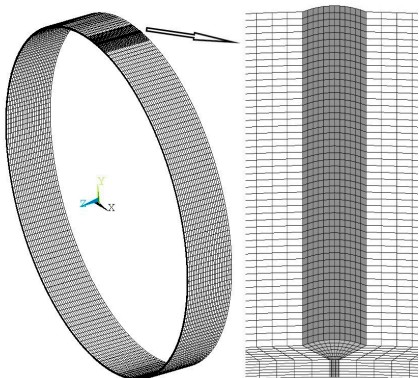

**Figure 2.** Model meshing diagram.

In the welding simulation, the inner and outer surfaces of the model were convective boundary conditions. The convection coefficient is 10 W·m$^{-2}$·K$^{-1}$, and the emissivity is 0.85. The initial ambient temperature was 20 °C. In the simulation process, the welding seam is divided into several layers, so that the depth of each layer is very small, and the Gaussian heat source can simulate this situation well [34]. So, in the thermal analysis, it is assumed that the heat input of the welding arc meets the Gaussian distribution. The expression of heat flow distribution is

$$q(r) = q_m e^{(-Kr^2)}, \tag{1}$$

where $q(r)$ is the heat flow density at the center of the heat source, $q_m$ is the maximum heat flow density at the center of the heat source, and $K$ is the heat concentration factor.

In actual welding, the base metal will melt, change from solid to liquid, and absorb energy, while the molten pool part will solidify, change from liquid to solid, and emit heat. Therefore, the latent heat of phase transition also has an influence on the temperature change, which must be considered when calculating the temperature change in the crystallization temperature range. Otherwise, it will cause a large deviation in the calculation result. For the treatment of latent heat of solid–liquid phase change, there are mainly equivalent specific heat methods: the enthalpy method and the temperature rise method. Among them, the equivalent specific heat method is relatively simple and practical and has high calculation efficiency. This method is used in the simulation of this article. The so-called equivalent specific heat method essentially converts the latent heat of crystallization into specific heat and adds it to the actual specific heat as the correction value of the specific heat of the alloy in the crystallization temperature range. The calculation formula is as follows:

$$C_e = \frac{\Delta Q}{\Delta T}, \tag{2}$$

$$C_p = C_e + C, \tag{3}$$

where $\Delta Q$ is the amount of heat change during a phase change (J/kg), $\Delta T$ is the amount of temperature change (°C), $C$ is the specific heat when latent heat is not considered (J/(kg·°C)), $C_e$ is the specific heat change (J/(kg·°C)), and $C_p$ is the equivalent specific heat (J/(kg·°C)).

The APDL language of ANSYS and the "method of killing activating elements" technology was used to simulate the multiple metal filling and heat source movement in the welding process. Welding process simulation is shown in Figure 3, where the arrow represents the welding sequence. Based on the actual welding process, under the cooperation of the rectangular coordinate system and cylindrical coordinate system, the cyclic dynamic thermal analysis and stress analysis were performed on the welding process of the welded joint structure, and the distribution of residual stress field after welding was obtained.

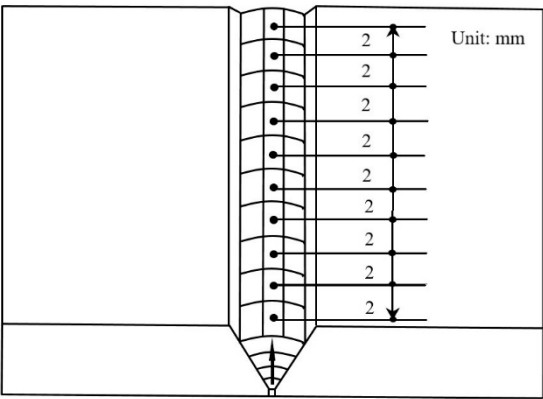

**Figure 3.** Schematic diagram of the welding process.

## 3. Results

### 3.1. Analysis of Temperature State in the Welding Process of Hoop Structure

Based on the welding process of a pressure vessel in practical engineering as a prototype, a finite element model of the welded hoop structure was established. In the simulation, the inner and outer cylinders were set to Q345R steel. The inner diameter of the laminate was 1800 mm, the thickness of the laminate was 12 mm, the weld width δ was 14 mm, and the length of the weldment was 1 m. The welding head adopted a V-shaped groove, and the angle of the weld groove was 40°. The welding current was 460 A, the arc voltage was 30 V, and the welding speed was 10 mm/s. The welding seam was divided into six lanes. Figure 4 is the temperature program of the second pass of welding. Figure 5 is the temperature cycle curve 1 mm away from the fusion line and 1 mm away from the lower surface of the upper plate. It can be known from Figure 5 that when the first layer of the weld is welded, the temperature around the weld reaches a maximum, and then the temperature is cooled to room temperature. As the distance from the heat source increases, the temperature of the node gradually decreases.

It can be seen from Figure 4 that with the activation of the unit, the heat source moves in a predetermined direction, and the temperature of the heat source and its surroundings decreases with increasing distance from the center of the heat source. It can be known from Figure 4b,c that with the movement of the welding heat source, the temperature distribution gradually reaches the quasi-steady-state stage, and the temperature field moves toward the welding line regularly until the end. It can be known from Figure 4d that as the heat source moves to the end of the weld, the welding is about to end. The temperature here changes rapidly, which will directly affect the size and distribution of the residual welding stress at the end of the weld.

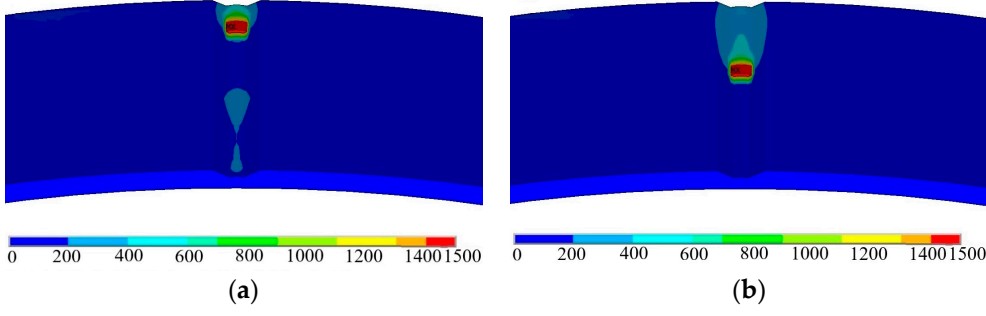

(a)　　　　　　　　　　　　　　　　　　　　　　(b)

**Figure 4.** *Cont.*

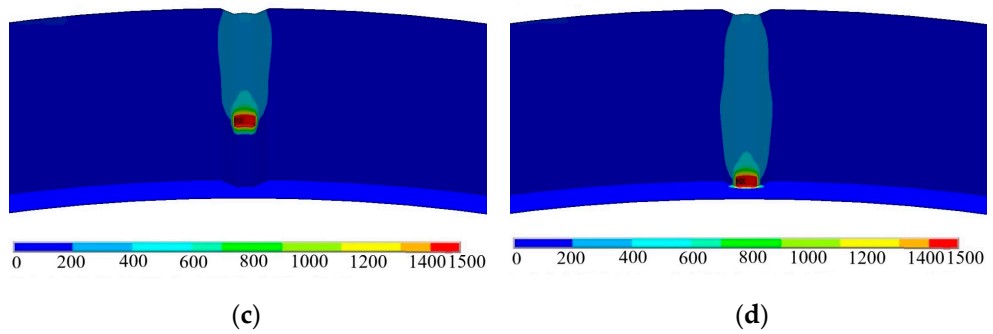

(**c**)　　　　　　　　　　　　　　　　　(**d**)

**Figure 4.** Temperature cloud diagram of the second layer weld welding. (**a**) t = 205 s; (**b**) t = 252 s; (**c**) t = 300 s; (**d**) t = 365 s.

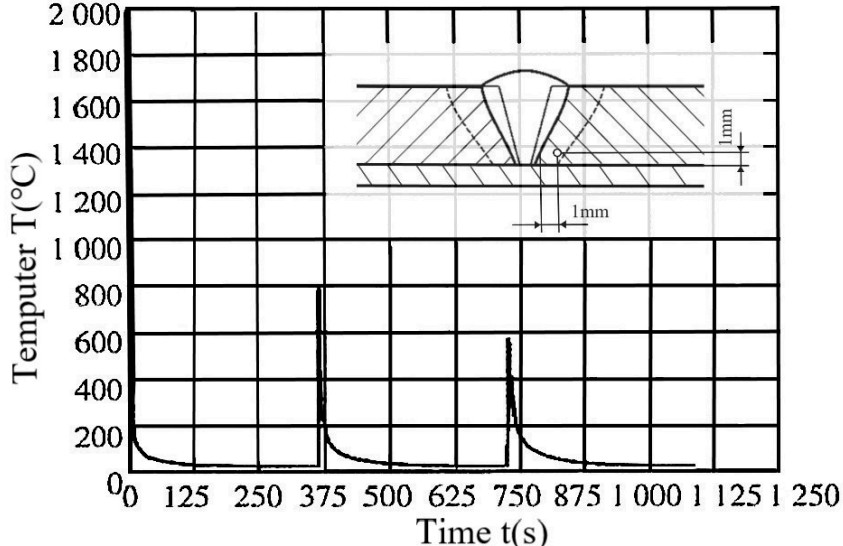

**Figure 5.** Temperature cycle curve at a point 1 mm from the fusion line and 1 mm below the lower surface of the laminate.

### 3.2. Analysis of the Distribution of Circumferential Residual Stress in the Hoop Structure

After welding and cooling, the distribution of the overall circumferential residual stress on the external surface of the structure is shown in Figure 6, where $\Phi$ represents the angle from the point of the laminate surface to the center of the weld, as shown in Figure 1. The top right corner of the picture is the stress cloud diagram of the weld and the surrounding area. The location of nodes used to study residual stress is shown in the black line in the figure. It can be seen from the figure that the maximum value of residual welding stress occurs on both sides of the interface between the weld and the weld plate. The stress value exceeds the yield strength of the base metal. With the increase of the distance from the weld, the residual stress on the outer surface of the hoop structure tends to zero gradually. Analysis of the stress distribution shows that during welding, the thermal expansion strain of the metal in the weld and near the seam area may be constrained by the surrounding colder metal, resulting in compressive plastic strain. In the process of welding cooling, the volume shrinkage of the weld and heat-affected zone produces tensile stress, so the tensile stress in the weld area shows an upward trend, while the tensile stress can only offset part of the compressive stress, so there is still residual compressive stress after welding. The structural characteristics of the hoop structure lead to a certain residual tensile stress in the cold metal region of the entire container during cooling shrinkage, and with the increase of the distance from the weld, the residual tensile stress decreases.

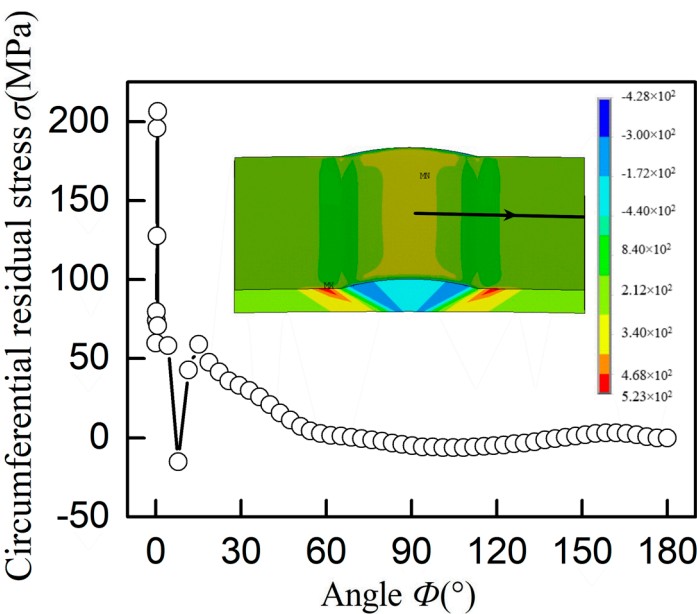

**Figure 6.** The distribution curve of circumferential residual stress in the hoop structure.

Figure 7 is the residual stress distribution diagram of the outer surface of the clamp structure at the value of $\Phi$ (Figure 1) between 0 and 20°. In Figure 7, A, B, C, D, and E respectively correspond to the welds and the surrounding nodes A, B, C, D, and E. It can be seen from the figure that the maximum residual tensile stress appears at the edge of the weld (point B), and the maximum residual compressive stress appears at the edge of the welded plate (point D). Welding is a fast and non-uniform thermal cycling process. When the welding heat source contacts the weld area, the temperature around the area rapidly rises to the melting point of the laminate material, causing the solder to fuse with the weld plate [35]. When the temperature is close to the melting point of the material, the elastic modulus of the steel decreases sharply. Hence, the melting zone and its edge regions are more plastic than in other regions. At this time, although the melting zone is thermally expanded, plastic deformation will occur, which will not cause high compressive stress. However, in the near-weld area, the material can still maintain the basic properties. During thermal expansion, it is restricted and squeezed by the surrounding low-temperature area, which causes local compression plastic deformation and compressive stress in the near-seam area, as shown in the CD section. During the cooling process of the weldment, the temperature around the weld decreases slowly, and the weld and heat-affected zone contract, which causes stretching to the far seam area, as shown in section DE of the figure. The plastic deformation of the metal in the CD segment due to local compression cannot be contracted freely, but it is stretched in the low-temperature zone. Hence, the increasing trend of compressive stress becomes slower. At the edge of the weld, it is in contact with the welding plate, with fast heat dissipation, fast temperature drops, and rapid shrinkage of the solder volume, resulting in high tensile stress between the units, as shown in section BC. The heat dissipation at the center of the weld is slow, so the temperature drops slowly, resulting in the slow volume shrinkage of the solder. Due to the symmetry of the structure, the tensile stress is small, as shown in the AB section of the figure. Therefore, the residual stress distribution is shown in Figure 7.

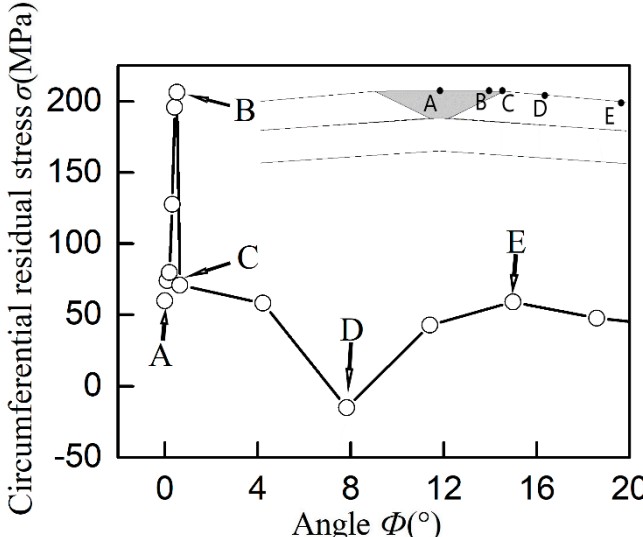

**Figure 7.** Distribution of circumferential residual stress around the weld of the hoop structure.

Similar to the overall circumferential residual stress distribution of the outer layer pipe, the circumferential residual stress of the inner layer pipe has a large variation near the weld, so the surrounding of the weld is taken as the main research object. Figure 8 is a diagram of the circumferential residual stress of the inner pipe with the $\Phi$ value between 0° and 10°. At the top right of the picture is the nephogram of circumferential residual stress around the inner tube weld. The black line in the figure is the location of the selected point, and the arrow is the direction of the selected point. As shown in the figure, due to the influence of the welding process, the maximum circumferential residual tensile stress of the inner tube is located at the center of the weld, which exceeds the yield limit of the material. As the distance from the center of the weld increases, the value of the circumferential residual stress rapidly attenuates; then, it gradually attenuates and tends to be stable.

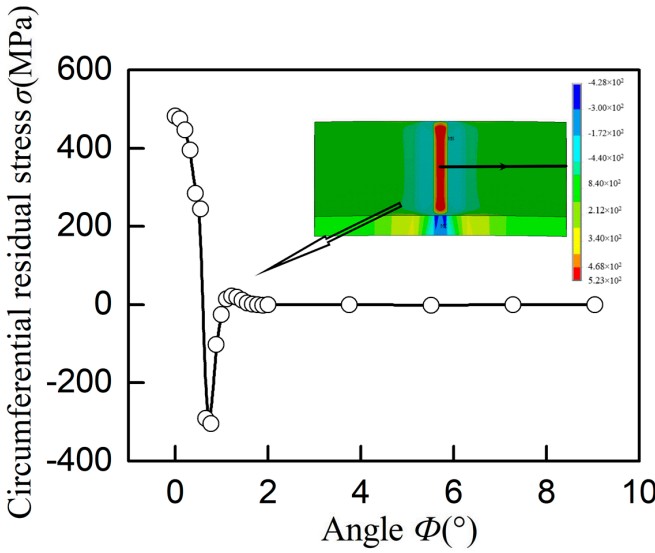

**Figure 8.** Distribution of circumferential residual stress in the weld around the inner tube weld.

## 4. Experimental Verification

In order to verify the correctness of the hoop structure simulation, the welding test under the same conditions was carried out, and the residual stress of the specimen was measured by the blind hole method. In the test, the inner and outer cylinder was Q345R steel. The inner diameter of the plywood

was 1800 mm, the thickness of the plywood was 12 mm, the width of the weld was 14 mm, and the length of the pipe is 1 m. A V groove shall be adopted for the welding joint, and the angle of the weld groove shall be 40°. The welding current was 460 A, the arc voltage was 30 V, the welding speed was 10 mm/s, and the diameter of the welding wire was 4 mm. During the welding process, electrode arc welding is used as the base, and submerged arc welding is used to fill and cover the surface.

Due to the large experimental materials, non-destructive residual stress detection is inconvenient. Therefore, the YC-III type static program-controlled resistance strain gauge and ZDL-II type residual stress drilling device were used to test the residual stress of the laminate by the blind hole method (Figure 9). After the laminate was wrapped and welded, in the stable area of the weld, strain gauges were respectively pasted in the $\Phi$ values of 2°, 5°, 10°, 15°, 45°, and 90° directions. Then, we used the drilling device to drill the blind hole with the hole diameter of 1.5 mm and the depth of 2.0 mm, and we measured the corresponding strain value. The residual stress value was calculated according to the principle of stress release. The calculation formula was:

$$\sigma_{1,2} = \frac{E(\varepsilon_1 + \varepsilon_2)}{4A} \pm \frac{E}{4B} \sqrt{(\varepsilon_1 - \varepsilon_2)^2 + [2\varepsilon_2 - (\varepsilon_1 + \varepsilon_2)]^2}, \tag{4}$$

$$\theta = \frac{1}{2} \text{tg}^{-1} \frac{2\varepsilon_2 - (\varepsilon_1 + \varepsilon_3)}{\varepsilon_3 - \varepsilon_1}. \tag{5}$$

Subsequently, the strain release systems *A* and *B* were modified with the shape-strain specific energy parameter, and the correction formula was [36]:

$$A(\mu\varepsilon/MPa) = \begin{cases} -0.341163 & S(10^{-8}) \leq 2.00417 \\ -0.005223 \times S - 0.330356 & S(10^{-8}) > 2.00417 \end{cases}, \tag{6}$$

$$B(\mu\varepsilon/MPa) = \begin{cases} -0.763261 & S(10^{-8}) \leq 2.00417 \\ -0.0219526 \times S - 0.719503 & S(10^{-8}) > 2.00417 \end{cases}. \tag{7}$$

The stresses at the measuring points in the x and y directions were:

$$\sigma_{x,y} = \frac{\sigma_1 + \sigma_2}{2} \pm \frac{\sigma_1 - \sigma_2}{2} \cos 2\theta. \tag{8}$$

The comparison between the test results and the finite element calculation results are shown in Table 3 and Figure 10. It is worth noting that because the experimental specimen is large, there would be a certain deviation in the residual stress measured at the corresponding point according to the angle. Therefore, the residual stress distribution trend was used as the verification standard here. It can be seen from the figure that the distribution trend of the residual stress on the outer surface of the hoop structure is basically consistent with the numerical simulation results, which can show the reliability of the finite element calculation.

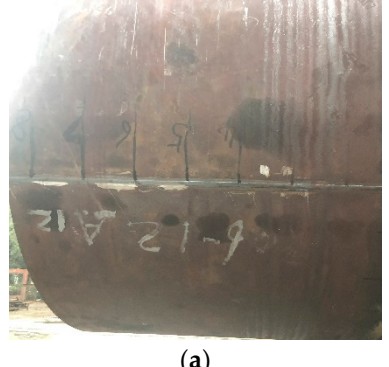
(a)

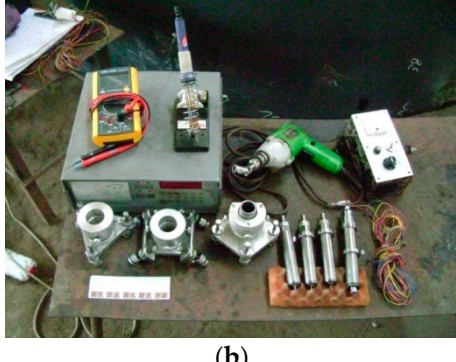
(b)

**Figure 9.** *Cont.*

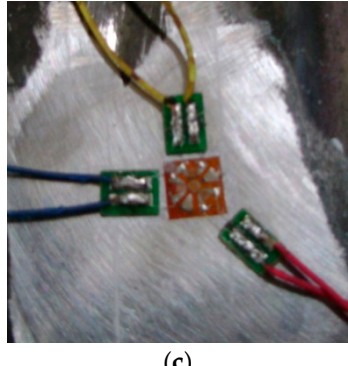
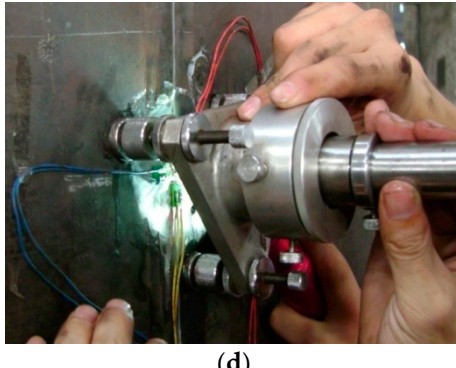

(**c**)                                                                          (**d**)

**Figure 9.** Experimental specimen and experiment apparatus. (**a**) Experimental specimen; (**b**) Drilling equipment and YC-III static stress measuring instrument; (**c**) TJ-120-1.5 strain gage rosette; (**d**) Drilling operation.

**Table 3.** Comparison between the results of the test and the results of the finite element calculation.

| Experiment Results $\sigma_E$ (MPa) | Numerical Simulation Results $\sigma_N$ (MPa) |
| --- | --- |
| 206.14 | 269.26 |
| 58.22 | 138.31 |
| −14.94 | −27.49 |
| 59.26 | 24.08 |
| 15.81 | 7.64 |
| −5 | 5.23 |

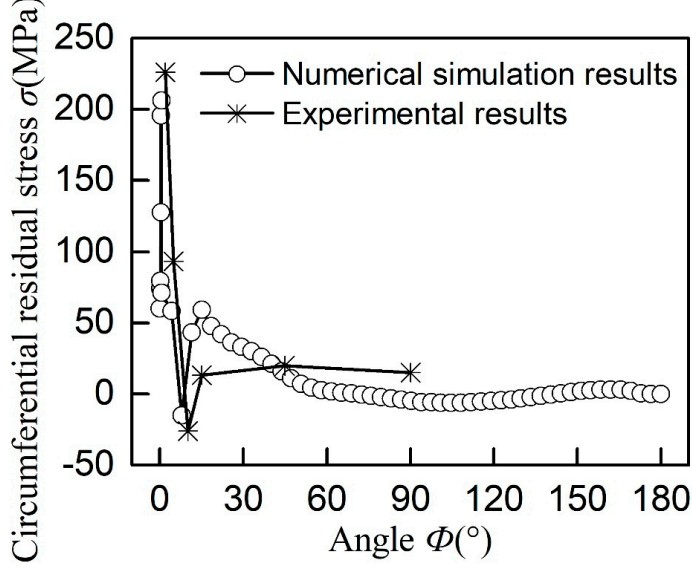

**Figure 10.** Comparison between the results of the test and the results of the finite element calculation.

## 5. Factors Affecting the Maximum Residual Stress of the Hoop Structure

The distribution of residual welding stress depends on several main factors, such as the structural dimensions, material properties, restraint conditions, heat input, number of weld passes, and welding sequence. These factors will affect the maximum welding residual stress on the laminate surface [16]. In this study, the following important factors were selected for research.

*5.1. Effect of the Radius on the Maximum Residual Stress of the Hoop Structure*

In order to study the influence of radius on the maximum residual stress of the welded hoop structure, the finite element simulation of a hoop structure with a different radius was carried out. The simulation results are shown in Figure 11. As the radius increases, the maximum residual stress of the hoop structure decreases. The reason may be that as the radius of the container increases, the curvature of the central heating area of the weldment decreases. The welding laminate tends to be flat, which reduces the heat dissipation area at the weld joint and reduces the residual stress caused by the uneven contraction of the hoop structure during the cooling shrinkage of the solder.

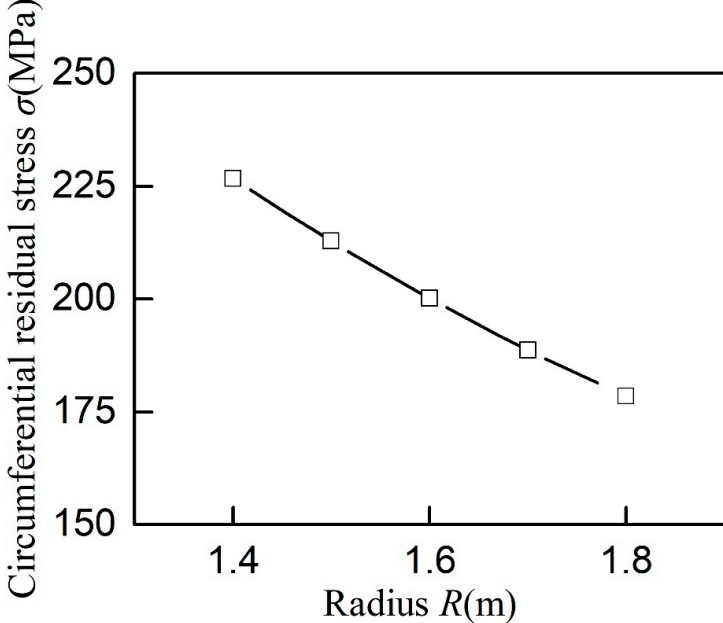

**Figure 11.** The distribution curve of maximum welding residual stress of the hoop structure with different radiuses.

*5.2. Effect of Welding Power and Welding Speed on the Maximum Residual Stress of the Hoop Structure*

The total heat input or internal heat source in arc welding is the product of arc power and process efficiency. In this simulation method, the correct heat input selection and modeling are essential to the outcome. It is essential to understand the concept of heat input and heat source efficiency, which varies between 65% and 85% for gas metal arc welding [19]. Since energy cannot be fully applied to the heating weldment, the power that is effective for heating welding is [37]:

$$Q = \eta Q_0 = \eta UI, \tag{9}$$

where $\eta$ is the thermal welding efficiency, $Q_0$ is the arc power (W), $U$ is the arc voltage (V), and $I$ is the welding current (A).

According to Formula (9), the influence of current and voltage on welding is proportional to the welding power. At the same time, at the same welding output power, the transient temperature field distribution caused by heat source movement will also affect the distribution of residual welding stress. In order to study the influence of welding power and welding speed on the maximum welding residual stress of the clamping structure, the welding process of the hoop structure under different welding powers and welding speeds was simulated. The influence curve of welding power and speed on the maximum welding residual stress is shown in Figure 12.

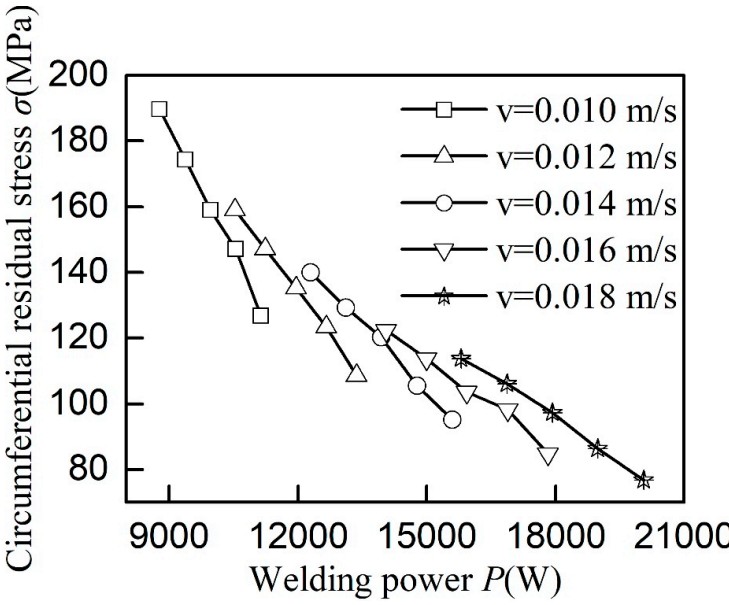

**Figure 12.** The distribution curve of maximum welding residual stress of the hoop structure under different welding powers and welding speeds.

It can be seen from the figure that as the welding energy increases, the maximum welding residual stress of the hoop structure decreases accordingly. It should be that with the rise of welding energy, the temperature of the solder and the melting zone increases, and the elastic modulus of the welding material decreases sharply. Compared with the surrounding cold metal, it can be regarded as a plastic material, so its ability to resist the extrusion in the surrounding low-temperature zone was relatively reduced, resulting in reduced compressive stress. In addition, with the increase of the temperature of the weld area in the melting state, the surrounding heating area became more substantial, and the overall heat dissipation time becomes longer, so the stress produced in the cooling process tended to decrease. Therefore, the increase of welding energy can effectively reduce the generation of residual stress on the surface of the laminate. As the welding speed increases, the maximum welding residual stress on the surface of the laminate decreases accordingly. This may be because of the addition of welding speed, which makes the temperature around the weld rise more evenly and can effectively change the uneven expansion caused by the irregular temperature change. However, in actual engineering, the increase in soldering speed will cause insufficient solder filling, so when considering the soldering rate, sufficient solder filling should be satisfied. Therefore, under the condition that the solder can be fully filled, the proper increase of the welding speed can effectively reduce the generation of residual welding stress of the hoop structure after the welding is completed.

*5.3. Effect of Weld Width and Thickness on the Maximum Residual Stress of the Hoop Structure*

In order to study the influence of weld width and laminate thickness on the maximum residual stress of the hoop structure, the finite element simulation of the welded clamp structure with different weld widths and the thickness of the hoop structure was performed. The maximum welding residual stress curve is shown in Figure 13. It can be seen from the figure that the maximum residual stress decreases with the increase of weld width and laminate thickness. The reason should be that with the increase of the weld width, the solder area will increase, resulting in a slower heat dissipation speed in the weld area, easing the residual stress caused by the sharp contraction of solder during the heat dissipation process, and at the same time, the increase of the stress area at the weld will reduce the extreme value of residual stress. When the width of the weld is constant, the amount of solder increases with the increase of the thickness of the laminate, which slows down the heat dissipation

process of the welding and reduces the residual tensile stress caused by the sharp contraction of the weld during cooling.

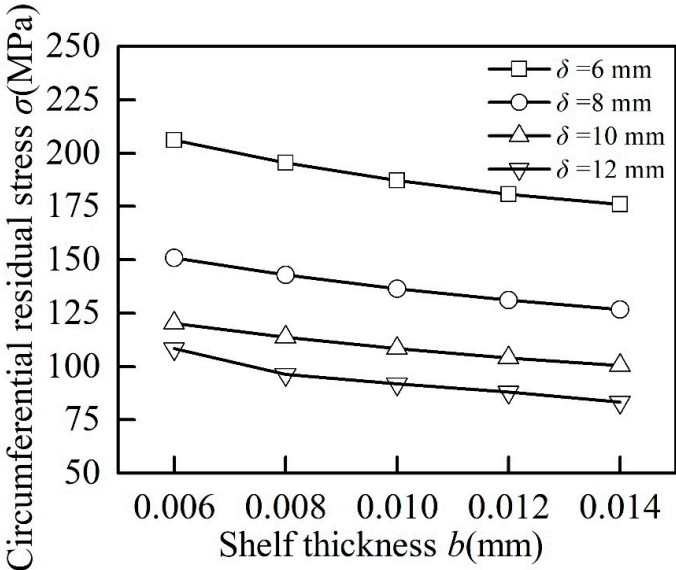

**Figure 13.** The distribution curve of maximum welding residual stress of the hoop structure under the different thickness values of the laminate.

## 6. Optimal Design of Process Parameters

According to the distribution of the maximum welding residual stress of the hoop structure under different welding parameters, the curve fitting of the maximum welding residual stress under different welding parameters is carried out, in order to obtain the best welding parameters to minimize the residual welding stress within the allowable parameters of the actual working conditions.

In this study, the maximum residual stress of the welded hoop structure was taken as the objective function. The welding speed, the thickness of the laminate, and the width of the weld were represented by the design variables $x_1$, $x_2$, and $x_3$ respectively. The reglm function of multiple linear regression analysis and generalized regression analysis were called in MATLAB to perform ternary one-time regression fitting on the parameters of Table 4. Table 5 shows the primary regression parameters. The fitting correlation coefficient is 0.9574, which indicates that the method is effective.

The corresponding $p$ values of the constant terms $x_1$, $x_2$, and $x_3$ are all less than 0.05, indicating that these terms in the regression equation are significant. The ternary linear regression equation for the maximum residual stress σ is:

$$F = 525.5378 - 13099.1667x_1 - 14305x_2 - 4121.1111x_3. \tag{10}$$

According to Formula (7), the residual welding stress of the hoop structure decreases with the increase of the thickness of the laminate, the width of the weld seam, and the welding speed when the actual working condition allows. According to China's current pressure vessel welding standards, the width of the butt welded joint (weld seam) should be determined according to the groove, and the form of the groove is determined by equipment and technology. According to the requirements of HG3129-1998, when the welding temperature is 1500 °C, the radius of the container is 1.8 m, the solder can be fully filled, the welding speed is 0.018 m/s, the width of the weld is 0.014 m, and the thickness is 0.014 m, the optimum solution exists. At this time, the maximum residual stress $\sigma_{min}$ = 31.89 MPa.

**Table 4.** Maximum residual stress value under different parameters.

| Data Number No | Welding Speed $v$ (m) | Shelf Thickness $\delta$ (m) | Weld Width $b$ (m) | Maximum Residual Stress $\sigma$ (MPa) |
|---|---|---|---|---|
| 1 | 0.01 | 0.008 | 0.008 | 260 |
| 2 | 0.01 | 0.008 | 0.01 | 247 |
| 3 | 0.01 | 0.008 | 0.012 | 237 |
| 4 | 0.01 | 0.01 | 0.008 | 218 |
| 5 | 0.01 | 0.01 | 0.01 | 208 |
| 6 | 0.01 | 0.01 | 0.012 | 200 |
| 7 | 0.01 | 0.012 | 0.008 | 189 |
| 8 | 0.01 | 0.012 | 0.01 | 182 |
| 9 | 0.01 | 0.012 | 0.012 | 175 |
| 10 | 0.012 | 0.008 | 0.008 | 223 |
| 11 | 0.012 | 0.008 | 0.01 | 212 |
| 12 | 0.012 | 0.008 | 0.012 | 203 |
| 13 | 0.012 | 0.01 | 0.008 | 176 |
| 14 | 0.012 | 0.01 | 0.01 | 168 |
| 15 | 0.012 | 0.01 | 0.012 | 161 |
| 16 | 0.012 | 0.012 | 0.008 | 163 |
| 17 | 0.012 | 0.012 | 0.01 | 156 |
| 18 | 0.012 | 0.012 | 0.012 | 150 |
| 19 | 0.014 | 0.008 | 0.008 | 196 |
| 20 | 0.014 | 0.008 | 0.01 | 186 |
| 21 | 0.014 | 0.008 | 0.012 | 178 |
| 22 | 0.014 | 0.01 | 0.008 | 165 |
| 23 | 0.014 | 0.01 | 0.01 | 157 |
| 24 | 0.014 | 0.01 | 0.012 | 150 |
| 25 | 0.014 | 0.012 | 0.008 | 144 |
| 26 | 0.014 | 0.012 | 0.01 | 137 |
| 27 | 0.014 | 0.012 | 0.012 | 132 |

**Table 5.** Parameter estimation of ternary regression equation.

| Variables | Constants | $x_1$ | $x_2$ | $x_3$ |
|---|---|---|---|---|
| estimated value | 525.5378 | −13099.1667 | −14305 | −4121.1111 |
| $p$-value | 0.0000 | 0.0000 | 0.0000 | 0.0001 |

## 7. Conclusions

This study investigated the residual stress of the hoop structure caused by the cooling shrinkage of the weld when the outer cylinder was wrapped and welded under the condition of the existing inner cylinder. In this study, the method of combining finite element simulation and experiment was used to analyze the welding forming process and welding circumferential residual stress of the hoop structure under the consideration of the friction between layers. Further study will research the influencing factors of the maximum welding residual stress of the hoop structure. The main research results in this study are concluded below:

(1) In the welding direction, the residual stress at the starting and ending arc changes significantly, and the residual stress at the center tends to be stable. In the circumferential direction of the pipe, the residual stress around the weld zone of the outer pipe changes significantly. The residual stress of the welded laminates has the highest value on the two sides, where the weld and the welded plate intersect. The maximum residual compressive stress appears at the edge of the welding plate. The maximum circumferential residual tensile stress of the inner tube is located at the center of the weld. With the increase of the distance from the center of the weld, the value of the external residual stress rapidly decays.

(2) The blind hole method was used to test the residual stress of the clamped structure to verify the correctness of the model. The finite element simulation analysis was performed on the

multilayered pressure vessel with different welding parameters. It can be seen that the maximum residual stress on the surface of the welding laminate shows a decreasing trend with the increase of welding energy, laminate thickness, weld width, welding speed, and vessel radius. Based on the overall multilayer clamped high-pressure vessel standard HG3129-1998 as a parameter, a three-dimensional first-order equation of the maximum residual stress on the welding speed, the thickness of the laminate, and the width of the weld seam is established according to numerical simulation results and optimized The optimal welding parameters under the actual working conditions were obtained, which provided a reference basis for the safe use and optimal design of the welding hoop structure.

**Author Contributions:** Conceptualization, W.M. and H.Z.; methodology, W.M.; software, Z.H.; validation, W.M., H.Z., and W.Z.; investigation, W.Z.; resources, F.X.; data curation, H.Z.; writing—original draft preparation, H.Z.; writing—review and editing, W.M.; visualization, W.Z.; supervision, W.Z.; project administration, C.Y.; funding acquisition, C.Y. All authors have read and agreed to the published version of the manuscript.

**Funding:** This research was funded by the National Natural Science Foundation of China (2019JJ50625), the Key Research and Development Plan of Hunan Province (2017WK2032), the Special project of innovative province construction in Xiangtan City (CG-TPZJ20191028), the Innovative Venture Technology Investment Project of Hunan Province (2018GK5028), the Key R&D Program of Hunan Province(2018WK2111), the S&T Innovation Platform and Talent Plan of Hunan Province (2017XK2048), and the Major Program of Science and Technology of the Hunan Province (2017SK1010).

**Conflicts of Interest:** The authors declare no conflict of interest.

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
