# Peer review of "Study on Residual Stress of Welded Hoop Structure"

_applsci, doi:10.3390/app10082838_

Round 1
Reviewer 1 Report
In the proposed manuscript, the authors examine the residual stresses when welding the rim using ABAQUS and ANSYS software. An attempt was made to compare the simulation results with the experiment. The reviewer's opinion has several disadvantages:
I suggest that the authors add a comparison of other programs to simulate welding processes (e.g. SYSWELD, Simufact Welding etc.).
There are many examples of the use of such programs, for example: DOI: 10.1007/978-3-319-96601-4_18, DOI: 10.1007/s40194-019-00718-z
I'm guessing that lines 138-140 do not bring relevant information to the article.
I suggest you transfer information about applicability ( Featured Application lines 11-16) to final conclusions.
page 3 line 102 - I will not agree with the statement that Young's modulus above 600 degrees does not have much influence on stress - after all, it is still half of its value (example in DOI: 10.1177/1687814015594313).
SYSWELD takes this value into account up to the melting point.
ANSYS does not take into account metallurgical changes, therefore forecasting the state of stress with the help of such a tool is quite risky. If it were that simple, SYSWELD, Simufact or Virfac programs would not have a place on the market.
The article proposes a heat source model with a Gaussian distribution. Apart from the fact that chapter 2.1 did not mention which method was welded (only on page 11 in line 292) and the selection of the heat source model was not justified, the method parameters were given only on the next page, still not mentioning the method. Much more convenient and better is e.g. the source in the shape of a double ellipsoid (Goldak), which is currently widely used in numerical analyzes. It did not show how or whether the heat source model was calibrated and validated.
The proposed model is very poorly described. Boundary conditions, i.e. fastening, heat dissipation - only this was mentioned, and since the whole article is based on the results of simulation and not testing of welded joints, this should be extended / supplemented.
The statement on page 5 line 150 - for a situation where the start of welding the second bead while the previous bead cools is incomprehensible.
"It can be seen from Figure 4 (a) that as the preset structure is long, when the second weld is started to weld, the previous weld has started to cool, and the temperature gradient around the welding decreases with the increase of the distance from the heat source. "
The temperature gradient will always be less as we move away from the heat source. Here, one would have to show whether heating by the previous bead does not significantly affect this gradient - e.g. by including the course of calculated thermal cycles at selected nodes.
Fig. 5 is illegible - I would give up the whole rim and replace it with an enlarged section.
"The maximum tensile stress can reach 532MPa, which exceeds the yield limit of the material, and the maximum compressive stress can reach 428MPa" - nothing comes from this sentence, i.e. there is no comment. The reported value of the maximum stress is questionable if we did not take into account metallurgical transformations. In addition, the statement that it exceeds Re - but the basic material and not the welded joint.
Fig. 6 is a medium readable - is enough space to add to his scheme and the same applied to the weld zone and HAZ.
The division into ABCDE nodes on page 6 is difficult to understood.
page 7 line 191 - "Welding is a fast and non-uniform thermal cycling process "- if the calculated thermal cycles are presented, then it can be said whether it really is" fast ". At 460A and 10 mm / s, it will not be such a sharp cycle, especially with pre-heating and multi-bead welding. Too far-reaching statement.
page 7 line 202 - "section De" - I think it's "section DE"
In general, all drawings from the simulation could be presented in a much more readable way and on a rather equal scale because e.g. 7 and 9 are two different sections. In addition, they are illegible at their present size. The same applies to figures 6, 8 and 10 - too small, illegible and strange that there are so few points.
If simulations have already been carried out and the Experimental verification chapter has been included - these data should be compared in a simple and visible way.
Figure 13 does it in a hardly legible way due to its form. Is this the only verification? Macroscopic photo of the welded joint and temperature distribution?
The results presented in chapter 5 are simulation results or real tests? My guess is that the simulation but the charts are not enough, it is good to supplement them also with the distributions of the analyzed quantities (even a few examples each).
Reviewer 2 Report
- Citations made in the text do not respect the template. The whole article must be revised from this point of view (R43, R48, R50 ...).
- Font size do not respect the template (Paragraph from R45 to R66; R67 to R72 ...). The whole article must be revised from this point of view.
- The introduction must be rewritten. There is no connection between all the data presented.
- The data are not coherent presented.
- Data verification is done only through a single experiment (v=10 mm/s) - it is insufficient.
Round 2
Reviewer 1 Report
The authors responded to substantive comments on the methodology and corrected the manuscript. However, I still found some editorial mistakes. I believe that the article can be published after applying these minor corrections.
Correct the references in accordance with the instructions of the journal for authors and complete some items whose bibliographic descriptions are incomplete (sometimes the number of the journal is missing, and sometimes even the name of the journal). For example, complete information available on the pages of magazines:
[4] Hashemi, J., Rasty, J., Li, S., and Tseng, A.A., "Integral Hydro-Bulge Forming of Single and Multi-Layered Spherical Pressure Vessels," ASME Transactions, Journal of Pressure Vessel Technology, Vol. 115, No. 3, August 1993, pp. 249-255.
[22] Kik T., Moravec J., Nováková I. (2018) Application of Numerical Simulations on 10GN2MFA Steel Multilayer Welding. In: Awrejcewicz J. (eds) Dynamical Systems in Applications. DSTA 2017. Springer Proceedings in Mathematics & Statistics, vol 249. Springer, Cham
[23] Sajek, A. (2019). Application of FEM simulation method in area of the dynamics of cooling AHSS steel with a complex hybrid welding process. Welding in the World, 63(4), 1065-1073.
Check references and standardize them as required by guide for authors.
Please change the symbol and unit notation in the whole article to the one suggested below (it looks more professional).
„ Yield stress σ/MPa ” change to „Yield stress σ [MPa]” - This applies to tables, drawings and charts (axis description) :
(table 1, table 2, fig. 5, fig. 6, fig 7, fig. 8, table 3, fig. 10, fig. 11, fig. 12, fig. 13, table 4)
International System of Units requires a space to be used to separate the unit symbol from the numerical value. This applies to text :
(line 106, line 151, fig. 4, lines 238 – 241, lines 249 – 250, lines 371-373)
Unlike degrees Celsius, the symbol of angular degrees is not separated by a space from the numerical value.
(line 98, line 195, line 224, line 248)
Reviewer 2 Report
Following the changes made to the article and the authors' responses, I recommend publishing it.
Author Response
The Comments and Suggestions of the Reviewer 2: Following the changes made to the article and the authors' responses, I recommend publishing it.